# An MeV Proton Irradiation Facility: DICE

**DOI:** 10.3390/ma17153646

**Published:** 2024-07-24

**Authors:** Sören Möller, Daniel Höschen, Wim Arnoldbik, Beata Tyburska-Pueschel

**Affiliations:** 1Aachen Ion Beams UG (Haftungsbeschränkt), 52066 Aachen, Germany; daniel.hoeschen@aachen-ion-beams.com; 2Dutch Institute for Fundamental Energy Research (DIFFER), 5612 AJ Eindhoven, The Netherlands; info@detect99.nl (W.A.); tyburska@differ.nl (B.T.-P.)

**Keywords:** materials, material analysis, irradiation, corrosion, nuclear power

## Abstract

Materials applied in nuclear environments such as fission or fusion power-plants face severe conditions. The irradiation by neutrons induces thermal loads and irradiation damage. Furthermore, coolants in contact with the materials induce corrosion, which is particularly challenging for liquid salts intended for the next generation of fission reactors. A new device (DICE) is installed at the 3.5 MV accelerator at DIFFER for the accelerated testing of such materials under combined irradiation and corrosion conditions. The DICE enables irradiation of samples at temperatures of up to 1050 K and in contact with liquid salts. An integrated shielding and a low power temperature control concept based on radiation cooling enables high-duty cycle application in a standard accelerator laboratory. Ion currents of up to 30 µA are possible with continuous irradiation. This work outlines the technical concept of the device and presents the first data.

## 1. Introduction

The investigation of material behavior under irradiation is an important aspect for the development of new nuclear power concepts. In operation, materials typically suffer strong neutron irradiation in the reactors resulting in transmutation and displacement damage. Those effects accumulate over the years of operation. When material problems, such as cracking or swelling, start to arise from this irradiation damage, the materials approach their end-of-life. For economic reasons, the materials should last from several years to decades, depending on their function inside the reactor. Logically, this would require decades of testing experiments in the real environment, a period that is too long for the dynamic development of urgently needed new power concepts.

Accelerators provide an option for the accelerated testing of materials under irradiation. The irradiation with MeV protons at high current densities can accelerate the irradiation testing by up to three orders of magnitude. At energies up to about 3 MeV, no significant nuclear activation takes place for most materials, only displacement damage [1]. This leaves out transmutation as a possible cause of damage, but also allows for a detailed investigation of the effects of irradiation in high-quality non-radioactive laboratories.

In a joint project, Aachen Ion Beams and DIFFER (Dutch Institute for Fundamental Energy Research) developed the DIFFER irradiation-corrosion experiment (DICE) setup as a prototype irradiation testing facility with high availability and low cost. DICE aims to investigate the response of materials to simultaneous proton irradiation and corrosion. A particular scientific goal is the investigation of the interaction of liquid salts with irradiated materials at high temperatures. Salts such as FLiNaK liquefy at 737 K and are intended for use as coolant liquids in Gen4 fission reactors. These salts can have corrosive effects [2,3,4], further shortening the material lifetime. Besides salts, other materials and coolants such as water and liquid metals can be investigated. The synergistic effects of combined irradiation and corrosion damage [5] can be investigated using the DICE. To our knowledge, the DICE is the second experiment of this kind [6] and the only one in Europe. However, the first one located at MIT operates at low currents and therefore low damage levels. To reach reactor-relevant damage levels a maximum ion-current density is required. The DICE is novel because it is designed to surpass existing devices in terms of ion-current density (up to 30 µA/cm^2^) by at least one order of magnitude [7] allowing for the simulation of about 1-year of molten salt reactor (MSR) performance within five days of operation. In the DICE, the beam current can be monitored continuously during irradiation. Additionally, it has safety systems installed allowing for the usage of radioactive salts like thorium chloride. These improvements make the DICE a unique experiment, enabling rapid material testing under concurrent harsh conditions such as irradiation, corrosion, and heat.

## 2. Vacuum Chamber and Beamline

The DICE setup combines a beamline with ion optics, an UHV chamber, a combined polyethylene and iron shielding, a resistive heater, and a sample holder. A magnetic quadrupole doublet enables the focusing and steering of the ion-beam [7] into the DICE. The chamber is connected to a beamline and a 3.5 MV Singletron ion accelerator. The advantage of this type of accelerator is its high beam stability with low ripple and high beam current [8]. The accelerator features an RF ion source delivering up to 75 µA for hydrogen and helium. When the source is filled with H_2_, it will produce an ion-beam composed of roughly 50% H^+^, 30% H_2_^+^, and 20% H_3_^+^. Since the DICE is installed on a beamline leading straight from the accelerator, a device for separating those H beams is required.

The solution to remove the H_2_^+^ and H_3_^+^ (H_2,3_^+^) beams is found by converting the first electrostatic Y steerer into a Wien (or velocity-) filter (WF) by adding a magnet, as shown in Figure 1. In our case, the H^+^ beam is the preferred beam and the H_2_^+^ beam has to be separated from that. Fortunately, the ions are light and there is a large √2 and √3 ratio in their velocities with respect to the velocity of H^+^, which makes it easy to separate the beams. The entrance flange of the 90° magnet vacuum chamber is used as a dump for the H_2,3_ ion beams. This flange has a horizontal letter-box shaped opening and the H_2,3_^+^ beams hit this flange under the center of the flange. To reduce the ionizing irradiation, this part of the flange is covered with a 1 mm tantalum shield. The distance between the WF and the flange is about 3.5 m. The effective length of the electrostatic steerer is calculated to be 24 cm. The pole length of the new magnet is limited by the electrical feedthrough of the steerer and amounts to 21 cm. However, this magnet is designed to have the poles around the 154 mm diameter wide housing of the steerer. This large pole distance increases the effective length of the magnetic field to 40 cm. With this configuration, an electrical field of 7.4 kV/cm and a magnetic field of 0.0188 T create a (vertical) separation of 3.0 and 4.5 cm, respectively, between the 3 MeV H^+^ and H_2,3_^+^ beams at a 3.5 m distance, with zero deflection for the preferred H^+^ beam. For the WF, 19 kV is needed to achieve this. Fortunately, 19 kV causes no risk of breakthrough with a plate distance of 25 mm. The aforementioned ratio between the currents of H^+^, H_2_^+^, and H_3_^+^ depends on the settings of the RF ion source. When the WF is switched on, the image of a beam profile monitor positioned upstream from the WF shows separate peaks for each of these ions. The ratio between hydrogen ions is estimated from the areas of these peaks.

Figure 2 shows the design of the DICE chamber and its supporting systems. The beamline is made from CF100 316 stainless steel (Hositrad Holland B.V., Hoevelaken, The Netherlands) vacuum tubing and has a length from sample to the accelerator of about 8 m. The beamline is equipped with a turbo- and fore-pump. An additional pumping station is close to the DICE chamber outside the shielding. The DICE pumping is equipped with a bypass for pump-down after the sample exchange. The DICE and beamline reach base pressures of 8 × 10^−7^ mbar. One Faraday cup with a combined scintillator imaging (FCBV—Faraday Cup Beam Viewer) is installed 1 m before the DICE for beam intensity and shape diagnostics, which can be moved out of the beam path during the irradiation experiments. A Rutherford-Backscattering Si-PIN detector is located in the beamline to monitor the beam current using the detector count rate during sample irradiation. The detector is located at a distance of 1 m to the sample, observing the sample at a 1° angle towards the beam axis. The beam passes an exchangeable aperture of up to 20 mm diameter and a fixed electrically isolated aperture of 18 ± 0.1 mm diameter (see Figure 3). This aperture is isolated against the other parts, enabling a separate current measurement or applying a secondary electron suppression bias for sample current measurement. The sample has a diameter of 15 mm diameter but the holder’s aperture decreased the available irradiation area to 13 mm diameter.

All apertures are equipped with tantalum shields for reducing the radiation levels in the laboratory when operating with higher energy protons. Nevertheless, the impact on the sample and stray protons can result in a significant dose rate during beam operation. The radiation dose originates mostly from neutrons in the MeV range produced by (p, n) reactions. Therefore, a shielding of 50 mm pure iron and 50 mm 5% boron-enriched polyethylene (PE) is mounted around the DICE chamber. The PE is the inner layer and reduces the dose rate through moderation and subsequent nuclear reactions of the neutrons with boron. This and possible activation produces photons, which are absorbed by the outer iron shielding. A part of the shielding is mounted on a sliding system in order to allow an easy access to the sample holder with minimal exposure times for personnel, in case of beam-off radiation exposure through decay radiation from nuclear activation products.

The sample chamber is made from 316 L steel and mechanically polished from the inside to reduce thermal losses from the hot sample holder. It features eight CF flanges for feedthroughs, detectors, sample observation, and sample access, together with an individually designed back plate to mount the sample holder, three thermocouples of type N with a 500 V isolation voltage and 1.5 K accuracy, electrical wirings for in-vacuum connections for sample heating, and a multi-pin feedthrough for temperature readings. Two flanges in the front provide direct view lines to the sample at an angle of 30°. Corresponding through-holes in the beam entrance and electrically isolated apertures enable a direct sample observation with visual and IR-cameras. The remaining flanges remain available for future additions.

## 3. Sample Holder

The sample holder as shown in Figure 3 is made from Inconel (Ni-based steel) in order to maximize the corrosion resistance and the possible sample exposure temperature. It holds a sample of up to 15 mm diameter with typical thicknesses in the order of 30 µm (range of 3 MeV protons in the metals), but higher thicknesses are possible. Four screws between the sample aperture and the salt chamber clamp the sample. The sample holder can be exchanged at low cost if other materials are required for new experimental challenges. Typically, the FLiNaK salt is operated at temperatures in the order of 1000 K, therefore one goal of the design is to allow high temperatures and low heating power. The sample holder is connected only through two screws at the entrance aperture to the sample chamber, strongly limiting its conductive temperature losses. Mostly radiative heat loss limits the sample temperature, which is further reduced through surface treatment and a heat shield.

The sample holder features an expansion volume on top in order to allow the salt to expand and degas upon heating. The open end of the expansion volume features a sinter filter for avoiding salt loss, but is otherwise open to the chamber vacuum. A clamp connects the expansion volume pipe to the sample chamber in order to cool it below the condensation point of the salt and avoid salt contamination of the chamber. A closed expansion volume or external salt circulation pump would have resulted in a pressure difference applied to the thin sample discs, increasing the risk of sample rupture, which is avoided by the implemented solution.

A direct contact of a temperature measurement to the sample is disregarded for the same reasons. The temperature is monitored via three thermo-elements of type N. The closest thermocouple (T2, see Figure 3) is about 5 mm away from the sample, measuring the Inconel holder in direct contact with the sample. The temperature in the sample center is therefore slightly higher than the measured value of the thermo-element. One more thermo-element is installed close to the sample as a backup (T1, partially seen on Figure 3). A resistive wire heater capable of heating up to 1300 K with up to 750 W is installed at the back of the sample holder together with another thermo-element monitoring (T3, not seen on Figure 3) the heater temperature. A feedback controller enables pre-heating and constant exposure to temperatures independent of the ion-beam current.

The coolant/liquid salt circulates only through gravity. The heat deposited by the ion-beam heats up the coolant behind the sample, which makes it move upwards into the larger coolant container. Here the salt cools down to the sample holder temperature. The surface quality and optional thermal shields enable an optimization of the sample holder temperature towards different scenarios. Surfaces that are more reflective will reduce the thermal radiation losses, which define the sample temperature due to the limited conduction losses of the sample holder.

## 4. Tests

A number of initial tests are conducted to demonstrate the device performance and important irradiation parameters. First the heating is explored with a heating power of up to 600 W. At 600 W, the sample holder heats up to a maximum of 1018 K within 110 min, as shown in Figure 4. The vacuum chamber temperature “T_case” increases only by up to 29 K above room temperature. The cooldown in the vacuum requires 24 h, since radiative cooling is largely ineffective at lower temperatures. By injecting a gas, e.g., N_2_ or Ar, during cooldown it can be reduced to 2 h.

Table 1 shows the connection between temperatures and heating power. A practical maximum temperature of 1047 K is found, so the sample temperature remains within the required safety limit in any case.

The test irradiation was performed using 2 MeV 10 µA H+ at the accelerator and 6.7 µA at the FCBV in front of the DICE. The beam was focused to its minimum size using the quadrupole doublet positioned before the chamber, resulting in the beam profile depicted in Figure 5. The beam profile is measured using the optical camera at the FCBV in front of the DICE. A full-width half-maximum beam size of 6.1 mm and 7.4 mm is found. The beam load of about 12 W resulted in a sample holder temperature increase of 16 K. A maximum of 30 µA ion-current at 2 MeV (=60 W beam load) is directed onto the sample by increasing the ion source output.

The procedure to adjust the beam to hit the sample is as follows. First, we minimize the beam size through adjustments of the quadrupole. Second, we move the beam using the steerer from left to right to find the minimum between two maxima of the floating aperture current signal. The ion-current impinging on the sample reduces the floating aperture current, resulting in two floating aperture current maxima around and central to the current minimum (the sample position) when steering the beam in a straight line across the sample holder. Once the maximum ratio of holder current divided by outer aperture current is found in the horizontal direction, a vertical steerer scan is conducted with the same aim. Repeating this one or two times enables a fine-tuning of the ideal beam position. In the optimal settings, an initial current of 6.7 µA at the F-cup is transmitted 98% through the 18 mm outer aperture onto the sample.

## 5. Conclusions

A new setup for material irradiation testing is in operation at DIFFER, named the DICE. The chamber combines robust technologies for enabling continuous long-term (>100 h) irradiation experiments with thin samples and liquid salts. The goal of increasing the ion-current density is achieved, enabling practical relevant damage levels. The approach of reduced thermal contacts and a low emissivity surface successfully allows relevant temperatures to be reached at low heating powers with a resistive heater, as proven by the heating and cooling investigations showing a non-linear increase of sample temperature with heating power. Slightly higher cooldown times are the only drawback, but the advantages of no active cooling being required and the surrounding PE shielding being able to remain in close contact with the vacuum chamber are achieved. A low heating power keeps the surroundings cool and provides generally safer operational conditions. Therefore, the DICE can be safely operated in regular laboratory environments, despite the potential hazards induced by the salts, the nuclear activation, and the high temperature.

The proton-irradiation of iron by 3 MeV with the current of 30 µA on a 1 cm^2^ spot would induce 0.9 dpa/24 h at the depth of 25 µm (calculated using SRIM-2013.00 [9] with Stoller et al. [10] method). This exceeds the damage rates of previous devices by about one order of magnitude and enables relevant studies for nuclear materials and corrosion. Defect creation at the DICE can be accelerated by approximately two orders of magnitude compared to neutron-induced defects, while the rate of corrosion remains unchanged. Consequently, experiments at the DICE do not provide a direct comparison to material performance in MSRs. However, the DICE serves as an effective screening tool, quickly identifying the best performing materials for subsequent testing in nuclear reactors.

## Figures and Tables

**Figure 1 materials-17-03646-f001:**
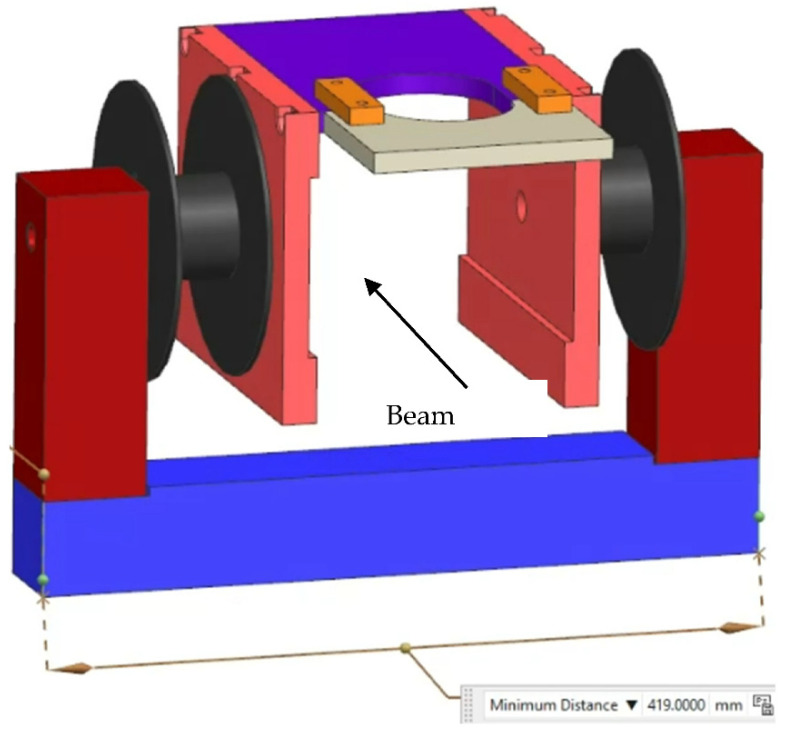
(color online): Schematic of the magnetic add-on for the electrostatic steerer to form a Wien filter used to separate and select H, H_2_ and H_3_ ions to enter the DICE. The coils are marked in black. The length of the blue bar is 41.9 cm.

**Figure 2 materials-17-03646-f002:**
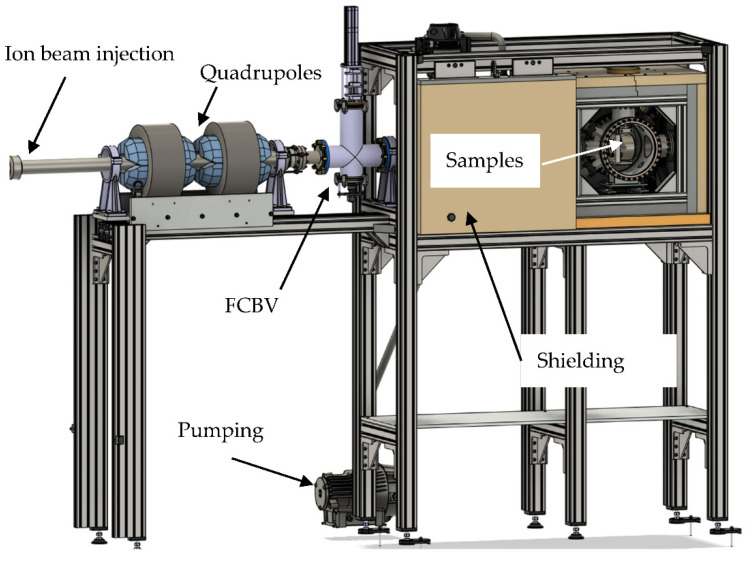
(color online): CAD drawings of the DICE setup—the vacuum chamber together with the shielding and beamline.

**Figure 3 materials-17-03646-f003:**
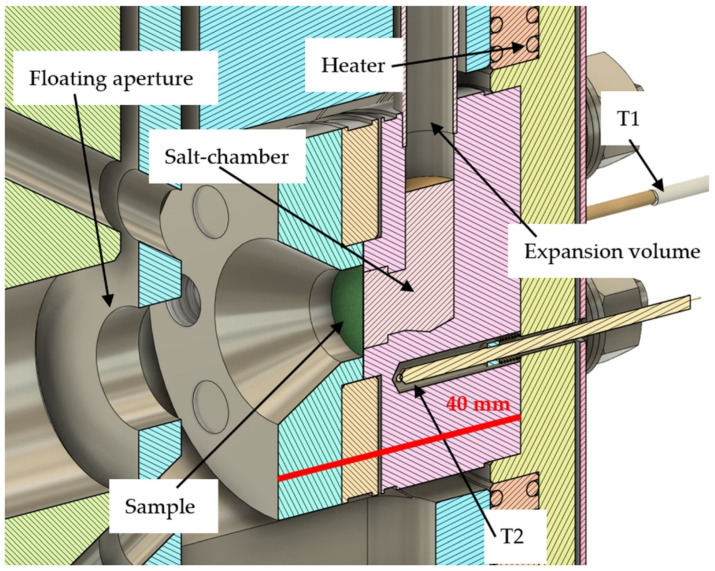
(color online): Details of the sample holder design. The ion-beam comes from the left and hits the green target disc. The liquid salt behind the sample can expand into an open volume. Two thermocouples (T1, T2) read the temperature nearby the sample. A heater is installed behind the salt chamber—its temperature is monitored by the T3 thermocouple (not visible). The scale bar of 40 mm is marked in red.

**Figure 4 materials-17-03646-f004:**
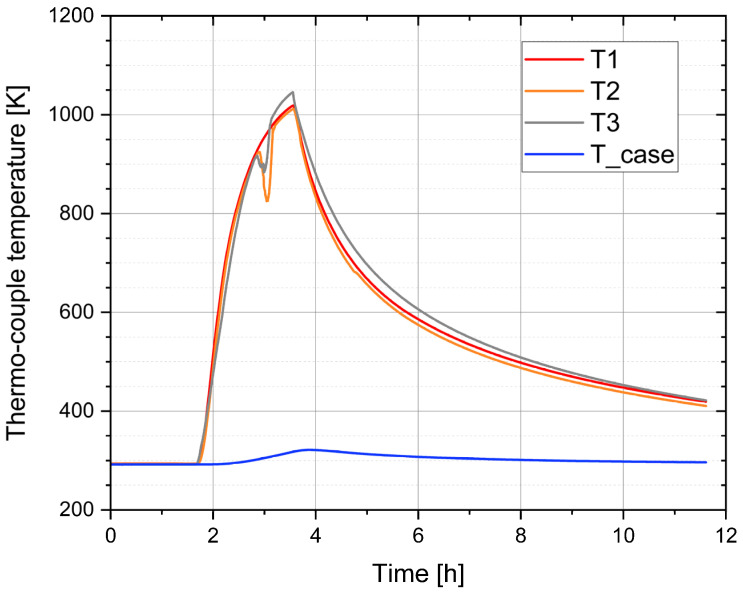
(color online): Time traces of the temperature evolution upon heating and cooling. T1 and T2 thermocouples are located close to the sample, T3 monitors the temperature of the heater, and T_case measures the temperature of the chamber. The signal from T2 and T3 at around 3 h was lost for a moment.

**Figure 5 materials-17-03646-f005:**
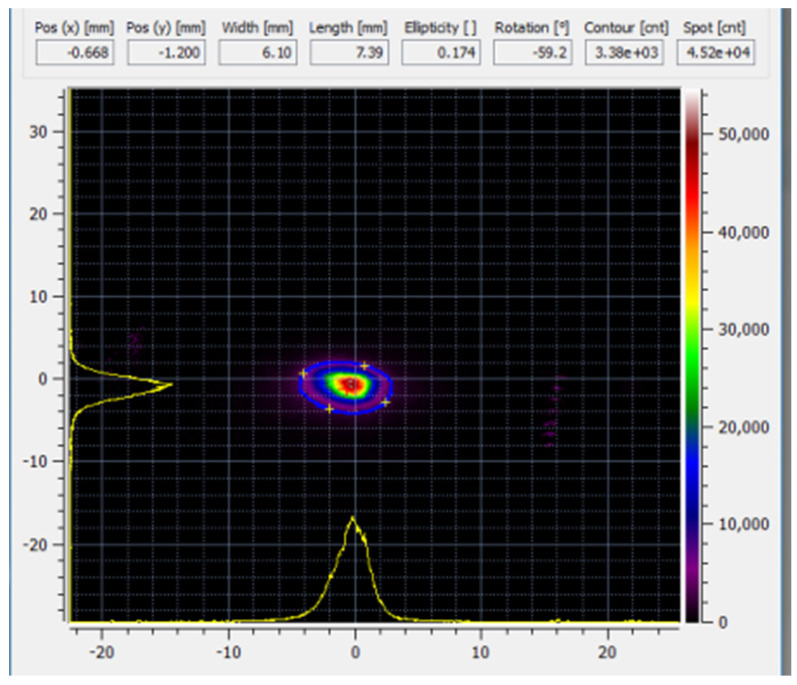
(color online): Optical image of the ion-beam dimensions on a quartz scintillator. Intensity is in arbitrary units with a maximum level of 64,000. The horizontal and vertical scales are in mm.

**Table 1 materials-17-03646-t001:** Measured sample temperatures during heating or irradiation. T3 is close to the heating wire. T1 is closest to the sample. T2 is between the sample and heater. A thermo-element “T_Case” is attached to the sample chamber outside.

Heating Power [W]	T1 [K]	T2 [K]	T3 [K]	T_Case [K]
0	293	293	293	293
12	309	309	309	293
100	700	692	694	297
170	800	796	800	298
600	1018	1011	1047	300

## Data Availability

Data are contained within the article.

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
