# Peer review of "An MeV Proton Irradiation Facility: DICE"

_materials, 2024, doi:10.3390/ma17153646_

Round 1

Reviewer 1 Report

Comments and Suggestions for Authors

PIF was designed primarily for single-event effect studies and characterization of electronic components and detectors for the space environment. The topic addressed is important and has several implications in particles accelerators. I found the manuscript interesting, However, it is not clear the real novelties in the present work. Hence, I would like to ask authors to better elucidate their novelties compared to what was done in literature.

Moreover, I think 1. references are too weak.

Some references are 

https://doi.org/10.1109/REDW.2000.896260

https://doi.org/10.1109/TNS.2019.2952003

IEEE TRANSACTIONS ON NUCLEAR SCIENCE, VOL. 52, NO. 6, DECEMBER 2005

https://doi.org/10.3390/instruments2040026

2. It is important to address properly what was done in literature and the novelties found in literature for readers. A table is not bad but for the present work, I don't think it will work due to the technical field addressed. However, maybe it will be a good idea to add a table regarding technical tools used. This is left for authors

Author Response

Comment 1

PIF was designed primarily for single-event effect studies and characterization of electronic components and detectors for the space environment. The topic addressed is important and has several implications in particles accelerators. I found the manuscript interesting, However, it is not clear the real novelties in the present work. Hence, I would like to ask authors to better elucidate their novelties compared to what was done in literature.

Moreover, I think 1. references are too weak.

Some references are 

https://doi.org/10.1109/REDW.2000.896260

https://doi.org/10.1109/TNS.2019.2952003

IEEE TRANSACTIONS ON NUCLEAR SCIENCE, VOL. 52, NO. 6, DECEMBER 2005

https://doi.org/10.3390/instruments2040026

Comment 2

It is important to address properly what was done in literature and the novelties found in literature for readers. A table is not bad but for the present work, I don't think it will work due to the technical field addressed. However, maybe it will be a good idea to add a table regarding technical tools used. This is left for authors

Response to the 1st review

We believe the first review does not pertain to our article. Our experiment focuses on simultaneous testing of material response to proton-irradiation and (salt)-corrosion, and not for single-event effect studies. The proposed literature is irrelevant to our device. The second comment does not align with any part of our article. However, we have followed the reviewer’s suggestion to better highlight the novelty of our device. These changes are marked in red.

Reviewer 2 Report

Comments and Suggestions for Authors

The work described in this paper is not particularly novel, but I think there is no obvious reason why the work should not be published.  However, I would suggest that the following few sub-edits be made to make the paper more immediately readable.

Lines 45-46:  should include specific numbers for ion-current densities (µA/cm^2 or similar).

Fig 1:  would be helpful to include some dimensions in the figure.

Line 67:  should be 1 mm, not one mm.

Line 74: '3, resp 4.5 cm' should read '3.0 and 4.5 cm respectively'.

Fig 2:  Fig. 2b seems to be missing.

Line 93:  replace 'floating' by 'electrically isolated' — initially I thought 'floating' meant mechanically floating.

Line 96:  an explanation of 'open diameter' is required — presumably it is the diameter of the sample behind the cone-shaped collimator that can be irradiated by the beam?

Fig 3:  in the figure caption it would be useful to give some numbers for physical dimensions, and to add something about the other two thermocouples T1 and T3, as otherwise T1, T2 and T3 appear rather out of the blue in Table 1.

Fig 4:  it is confusing that the temperature scale is in °C whereas elsewhere temperatures are in °K — temperatures should be all in the same units.

Fig 5:  the caption should say that the horizontal scale is in mm.

In addition, the authors might like to add a word or two about whether 30-µm-thick samples of materials and the much thicker samples of the same materials encountered in a real reactor are likely to respond in exactly the same way to irradiation and corrosion, and also about whether the radial gradient of the number of DPAs produced by such a small beam spot is significant.

And are the DPAs the usual NRT DPAs, or other sorts of DPAs?  For example ARC-DPAs (https://www-nds.iaea.org/public/download-endf/DXS/Displacement_XS/DXS.(2018)/)?

Recommendation:  publish after minor revisions.

Comments on the Quality of English Language

A few sub-edits have been suggested, otherwise fine.

Author Response

Comment 1

The work described in this paper is not particularly novel, but I think there is no obvious reason why the work should not be published.  However, I would suggest that the following few sub-edits be made to make the paper more immediately readable.

Response 1

To highlight the novelty of DICE we added the following text marked in red:

o our knowledge, DICE is the second experiment of this kind [6] and the only one in Europe. However, the first one located at MIT operates at low currents and therefore low damage levels. For reaching reactor relevant damage levels a maximum ion-current density is required. DICE is novel because it is designed to surpass existing device in terms of ion-current density (up to 30 µA/cm2) by at least one order of magnitude [7] allowing simulation of about 1-year molten salt reactor (MSR) performance within 5 days of operation. IN DICE, the beam current can be monitored continuously during irradiation. Additionally, it has safety systems installed allowing usage of radioactive salts like thorium chloride. These improvements make DICE a unique experiment, enabling rapid material testing under concurrent harsh conditions such as irradiation, corrosion, and heat.

Comment 2

Lines 45-46:  should include specific numbers for ion-current densities (µA/cm^2 or similar).

Response 2

This has been added to the text, in blue: (up to 30 µA/cm2)

Comment 3

Fig 1:  would be helpful to include some dimensions in the figure.

Response 3

A red bar is added to the figure and the following text in blue: The length of the blue bar is 41.9 cm.

Comment 4

Line 67:  should be 1 mm, not one mm.

Response 4

It has been changed to 1

Comment 5

Line 74: '3, resp 4.5 cm' should read '3.0 and 4.5 cm respectively'.

Response 5

Changed to: 3.0 and 4.5 cm respectively

Comment 6

Fig 2:  Fig. 2b seems to be missing.

Response 6

Indeed yes. We removed the corresponding text from the caption.

Comment 7

Line 93:  replace 'floating' by 'electrically isolated' — initially I thought 'floating' meant mechanically floating.

Response 7

This has been changed according to the reviewer’s suggestion

The beam passes an exchangeable aperture of up to 20 mm diameter and a fixed electrically isolated aperture of 18±0.1 mm diameter

Comment 8

Line 96:  an explanation of 'open diameter' is required — presumably it is the diameter of the sample behind the cone-shaped collimator that can be irradiated by the beam?

Response 8

This has been explained better as: The sample has a diameter of 15 mm diameter but holder’s aperture decreased the available irradiation area to 13 mm diameter.

Comment 9

Fig 3:  in the figure caption it would be useful to give some numbers for physical dimensions, and to add something about the other two thermocouples T1 and T3, as otherwise T1, T2 and T3 appear rather out of the blue in Table 1.

Response 9

This has been adjusted as follows: Details of the sample holder design. The ion beam comes from the left and hits the green target disc. The liquid salt behind the sample can expand into an open volume. Two thermo-couples (T1, T2) read the temperature nearby the sample. A heater is installed behind the salt chamber – its temperature is monitored by T3 thermo-couple (not visible). The scale bar of 40 mm is marked in red.

Comment 10

Fig 4:  it is confusing that the temperature scale is in °C whereas elsewhere temperatures are in °K — temperatures should be all in the same units.

Response 10

The figure has been replotted: On y-axis thermo-couple temperature in [K] and on the x-axis time in hours. Also we changed the figure caption to: Figure 4 (color online): Time-traces of the temperature evolution upon heating and cooling. T1 and T2 thermocouples are located close to the sample, T3 monitors the temperature of the heater, and T_case measures the temperature of the chamber. The signal from T2 and T3 at around 3 hours was lost for a moment.

Comment 11

Fig 5:  the caption should say that the horizontal scale is in mm.

Response 11

The Figure 5 caption is now:

Figure 5 (color online): Optical image of the ion beam dimensions on quartz scintillator. Intensity in arbitrary units with a maximum level of 64,000. The horizontal and vertical scales are in mm.

Comment 12

In addition, the authors might like to add a word or two about whether 30-µm-thick samples of materials and the much thicker samples of the same materials encountered in a real reactor are likely to respond in exactly the same way to irradiation and corrosion, and also about whether the radial gradient of the number of DPAs produced by such a small beam spot is significant.

And are the DPAs the usual NRT DPAs, or other sorts of DPAs?  For example ARC-DPAs (https://www-nds.iaea.org/public/download-endf/DXS/Displacement_XS/DXS.(2018)/)?

Response 12

We added the following text with additional references

3 MeV proton-irradiation of iron with the current of 30 µA on a 1 cm2 spot would induce 0.9 dpa/24h at the depth of 25 µm (calculated using SRIM-2013.00 [9] with Stoller et al. [10] method). This exceeds the damage rates of previous devices by about one order of magnitude and enables relevant studies for nuclear materials and corrosion. Defect creation at DICE can be accelerated by approximately two orders of magnitude compared to neutron-induced defects, while the rate of corrosion remains unchanged. Consequently, experiments at DICE do not provide a direct comparison to material performance in MSRs. However, DICE serves as an effective screening tool, quickly identifying the best performing materials for subsequent testing in nuclear reactors.

Reviewer 3 Report

Comments and Suggestions for Authors

see attachment

Comments on the Quality of English Language

/

Reviewer 4 Report

Comments and Suggestions for Authors

This paper describes the DICE setup. DICE is a new device that is installed at the 3.5 MV accelerator DIFFER. DICE enables the testing of the materials under combined irradiation and corrosion conditions. 

1. The authors should define all acronyms. For example, what DIFFER stands for?

 2. DICE aims at investigating materials for nuclear applications through proton beams of up to 3 MeV provided by a DC accelerator. It would be nice to get more information about the accelerator.

3. The paper only has 4 References, the Reference list should be much more extensive and more detailed research on previous work in this field should be done.

4. The motivation for building DICE as well as the state-of-the art in the field should be more explained. The device itself is nicely presented.

5. Please rephrase the sentences: A first irradiation is conducted with 2 MeV 10 µA H+ at the accelerator delivered  181 µA at the FCBV in front of DICE. Focusing this beam using the quadrupole doublet installed in front of the chamber to minimum size results in a beam profile shown in Figure  5.

6. What were the beam dimensions depicted in Figure 5? Figure 5 should be exported from the programme and properly shown (so that we do not have a screenshot).

7. The Conclusion section should be more thorough, it feels rushed up.

Comments on the Quality of English Language

Minor English editing is needed.

Author Response

Comment 1

  1. The authors should define all acronyms. For example, what DIFFER stands for?

Response 1

That has been done for DIFFER and FCBV (marked in green). Common acronyms like UHV and IR remain without extra definition.

Comment 2

  1. DICE aims at investigating materials for nuclear applications through proton beams of up to 3 MeV provided by a DC accelerator. It would be nice to get more information about the accelerator.

Response 2

Reference [4] gives full details of the accelerator.

  1. J. W. Mous, R. G. Haitsma, T. Butz, R.-H. Flagmeyer, D. Lehmann, and J. Vogt, ‘The novel ultrastable HVEE 3.5 MV SingletronTM accelerator for nanoprobe applications’, Nuclear Instruments and Methods in Physics Research Section B: Beam Interactions with Materials and Atoms, vol. 130, no. 1, pp. 31–36, Jul. 1997, doi: 10.1016/S0168-583X(97)00186-9.

Comment 3

  1. The paper only has 4 References, the Reference list should be much more extensive and more detailed research on previous work in this field should be done.

Response 3

DICE is the 2nd device of this kind in the world. There is simply no literature on this matter. However, we added some references about molten salt corrosion.

Comment 4

  1. The motivation for building DICE as well as the state-of-the art in the field should be more explained. The device itself is nicely presented.

Response 4

We added the following text, which shows how unique DICE is:

To our knowledge, DICE is the second experiment of this kind [6] and the only one in Europe. However, the first one located at MIT operates at low currents and therefore low damage levels. For reaching reactor relevant damage levels a maximum ion-current density is required. DICE is novel because it is designed to surpass existing device in terms of ion-current density (up to 30 µA/cm2) by at least one order of magnitude [7] allowing simulation of about 1-year molten salt reactor (MSR) performance within 5 days of operation. IN DICE, the beam current can be monitored continuously during irradiation. Additionally, it has safety systems installed allowing usage of radioactive salts like thorium chloride. These improvements make DICE a unique experiment, enabling rapid material testing under concurrent harsh conditions such as irradiation, corrosion, and heat.

Comment 5

  1. Please rephrase the sentences: A first irradiation is conducted with 2 MeV 10 µA H+ at the accelerator delivered  181 µA at the FCBV in front of DICE. Focusing this beam using the quadrupole doublet installed in front of the chamber to minimum size results in a beam profile shown in Figure  5.

Response 5

We rephrased those sentences as follows: The test irradiation was performed using 2 MeV 10 µA H+ at the accelerator and 6.7 µA at the FCBV in front of DICE. The beam was focused to its minimum size using the quadrupole doublet positioned before the chamber, resulting in the beam profile depicted in Figure 5.

Comment 6

  1. What were the beam dimensions depicted in Figure 5? Figure 5 should be exported from the programme and properly shown (so that we do not have a screenshot).

Response 6

We added to the Figure 5 caption: The horizontal and vertical scales are in mm.

Unfortunately, the program has no option of exporting that data and we can only provide a screenshot.

Comment 7

  1. The Conclusion section should be more thorough, it feels rushed up.

Response 7

We added the following text:

3 MeV proton-irradiation of iron with the current of 30 µA on a 1 cm2 spot would induce 0.9 dpa/24h at the depth of 25 µm (calculated using SRIM-2013.00 [9] with Stoller et al. [10] method). This exceeds the damage rates of previous devices by about one order of magnitude and enables relevant studies for nuclear materials and corrosion. Defect creation at DICE can be accelerated by approximately two orders of magnitude compared to neutron-induced defects, while the rate of corrosion remains unchanged. Consequently, experiments at DICE do not provide a direct comparison to material performance in MSRs. However, DICE serves as an effective screening tool, quickly identifying the best performing materials for subsequent testing in nuclear reactors.

Round 2

Reviewer 4 Report

Comments and Suggestions for Authors

The authors have properly answered all the questions and comments raised. I would therefore suggest accepting the paper in its present form.